# Relationship between Microbial Composition of Sourdough and Texture, Volatile Compounds of Chinese Steamed Bread

**DOI:** 10.3390/foods11131908

**Published:** 2022-06-27

**Authors:** Lili Fu, Adriana Nowak, Hongfei Zhao, Bolin Zhang

**Affiliations:** 1Department of Environmental Biotechnology, Faculty of Biotechnology and Food Sciences, Lodz University of Technology, Wolczanska 171/173, 90-530 Lodz, Poland; adriana.nowak@p.lodz.pl; 2Department of Food Science and Engineering, College of Biological Sciences and Technology, Beijing Forestry University, Beijing 100083, China; zhaohf518@163.com

**Keywords:** sourdough, microbiota, lactic acid bacteria, flavor, texture, Northwestern China

## Abstract

The objective of this work was to explore the relationship between the microbial communities of sourdoughs collected from the Xinjiang and Gansu areas of China and the quality of steamed bread. Compared to yeast-based steamed bread, sourdough-based steamed bread is superior in terms of its hardness, adhesiveness, flexibility, and chewiness. It is rich in flavor compounds, but a significant difference in volatile flavor substances was observed between the two sourdoughs. A total of 19 strains of lactic acid bacteria (LAB) were isolated from the Gansu sourdough sample, in which *Lactiplantibacillus plantarum* and *Pediococcus pentosaceus* were the dominant species, accounting for 42.11% and 36.84%, respectively. A total of 16 strains of LAB were isolated from the Xinjiang sourdough sample, in which *Lactiplantibacillus plantarum* was the dominant species, accounting for 75%. High-throughput sequencing further confirmed these results. Clearly, the species diversity of Gansu sourdough was higher. The volatile profiles of the sourdoughs were similar, but differences in the individual volatile compounds were detected between the sourdoughs of the Gansu and Xinjiang regions. These results point out that the differences in the microbiota and the dominant strains lead to differences in the quality of sourdoughs from region to region. This investigation offers promising guidance on improving the quality of traditional steamed bread by adjusting the microorganisms in sourdough.

## 1. Introduction

Sourdough is a traditional dough starter. It is a mixture of flour and water, spontaneously fermented by lactic acid bacteria and yeasts, with acidification and fermentable capacities [1,2]. Extensive research has shown that the main microorganisms in the sourdough are yeasts and lactic acid bacteria (LAB), among which LAB plays an important role in the flavor and texture of sourdough [3,4,5]. To date, more than 70 species of LAB have been isolated from sourdough regarding the composition of LAB, including mostly *Lactobacillus* spp. as well as *Leuconostoc* sp., *Weissella* sp., *Pediococcus* sp., and *Enterococcus* sp. [6,7,8]. Data from several studies suggest that the differences in volatile substances were mainly attributed to the metabolism and enzyme conversion by different LAB [9,10]. Amino acids can be converted into alcohols, acids, and esters during fermentation, obtained from amino acid transaminase or deaminase [11,12]. Homo- and heterofermentative metabolism of LAB differ concerning the flavor formation in bread and enable the volatile substances produced by LAB, which are species-specific. [13,14,15,16]. Some research indicates that volatile substances vary within different species of LAB. Previous research has established that the microorganisms present in sourdough can be sourced from ingredients or the baking environment and are typically consistent over time. This variation is manifested in the flavor of the bread [17,18,19]. Of particular concern is the surface microbiome of bakers, which may be a source of yeast and bacteria contamination in the bread [20]. Thus, selected specific microbiota as starter cultures play a vital role in developing sourdough bread with specific flavor properties and quality, especially in the slow industrialization of Chinese steamed bread.

Sourdough is a product having regional characteristics. It is a unique food ecosystem that selects LAB adapted to their environment and harbors dominant species [5]. Several studies analyzing the microbial communities in sourdough collected from different regions of China have shown dramatic differences in the dominant strains [21,22]. *Leuconostoc citreum* HO12 and *Weissella koreensis* HO20 isolated from kimchi were assessed as starter cultures in the making of sourdough bread [16]. Four LAB strains were tested for the preparation and propagation of sourdoughs to produce a typical bread at the industrial level [23]. It was confirmed that dominant strains played a significant role in benefiting the quality of sourdough, improving the mouth-feel of bread. A clear understanding of the interaction between the dominant strains of sourdoughs and the quality of steamed bread will benefit the practice. Thus, it is essential to identify the composition of sourdough, selecting the dominant strains as starter cultures for promoting the quality of Chinese steamed bread.

Chinese steamed bread is a widely consumed food, representing about 40% of the wheat consumption in China, especially in Northwestern China [24]. Local people in Northwestern China, including Gansu, Xinjiang, Ningxia, Qinghai, and Shaanxi areas where high-quality and high-yielding wheat is produced, enjoy sourdough-based steamed buns daily [25]. The Uyghurs and Hui Minorities specifically have preserved their own dietary habits and specific lifestyles, developing a specific dietary culture in Northwestern China [26]. However, few reports dealing with the linking of sourdoughs from Northwestern China to their steamed-bread quality have been presented [21,27]. Due to the special dietary characteristics, geographical environment, and huge market in Northwest China, clarifying the relationship between the microbial communities of sourdough in Northwest China and the formation of texture and volatile compounds in steamed bread is of great significance for processing the staple food.

The objective of this study was to investigate the relationship between the microbial communities of sourdough obtained from Xinjiang and Gansu areas and the texture and volatile compounds of corresponding Chinese steamed bread. This work is trying to elucidate the connection of the dominant LAB strains with the traditional Chinese steamed bread’s quality by adjusting microorganisms in sourdough.

## 2. Materials and Methods

### 2.1. Sourdough Sampling

The sourdoughs analyzed in this study were collected from different families located in the Provinces of Xinjiang (XS) and Gansu (GS) of China, which produce sourdough bread without any added salt. The selected families have been making Chinese steamed bread (CSB) with sourdough every day for more than 10 years. Three sourdoughs were collected from each family. All samples were kept at a low temperature (4 °C) during transportation and storage.

### 2.2. Chinese Steamed Bread Making Using Sourdough

To evaluate the quality of the steamed bread made using sourdoughs from different areas, yeast CSB was added as blank control. The recipes of yeast CSB (WSB), Gansu sourdough bread (GSB), and Xinjiang sourdough bread (XSB) are shown in Table 1.

Firstly, the sourdough, basically classified as type I sourdough, was activated with sourdough and sterile water in the mass ratio of 2:3 and fermented at 25 °C for 9 to 24 h to obtain a mature sourdough. Secondly, the dough was prepared as described by Xi et al. (2020) with some modifications (see recipe in Table 1) [28]. The dough was prepared in a spiral mixer (JHMZ-200, Beijing Oriental Fude Technology Development Co., Ltd. Beijing, China) by mixing yeast/sourdough with flour and water for 10 min to smooth the surface of the dough. Next, the dough was divided into 50 g/piece and was placed in an incubator for fermentation at 37 °C with 80% relative humidity for 4 h [22]. The proofed dough was steamed for 20 min. Each dough treatment was processed in triplicate.

### 2.3. Isolation of LAB in Sourdough

An amount of twenty-five grams of each sourdough sample was aseptically taken, transferred into a conical flask, and homogenized with 225 mL of sterile water. Appropriate decimal dilutions of the samples were prepared using the same medium. From each dilution, 0.1 mL was inoculated on De Man, Rogosa, and Sharpe (MRS) agar medium (Shanghai Yaji Biotechnology Co., Ltd. Shanghai, China) containing CaCO_3_ using the plate coating method. Plates were incubated at 37 °C for 48 h to produce typical colonies. Subsequently, colonies with different morphologies and soluble transparent calcium circles were selected and streaked onto MRS agar for purification. After 48 h of incubation at 37 °C, pure isolates were subjected to the bacterial count test and Gram-stained. Only colonies with gram-positive cocci or rods were stored with the vacuum freeze-drying preservation method at −80 °C.

Pure isolates were grown in an MRS medium at 37 °C for 24 h and sequenced with 16S rDNA sequencing. The method for identifying LAB isolates is described by Fujimoto et al. [29]. Sequencing was carried out by Sangon Biotech (Shanghai, China). Sequencing was compared with the GenBank database using the BLAST program (http://www.ncbi.nih.gov/BLAST/, 28 December 2019) [30]. An unknown isolate in the database had a nearest neighbor exhibiting the highest similarity score of ≥97%.

### 2.4. Screening for Optimal LAB

#### 2.4.1. Determination of the Growth Capacity of LAB

Strains were grown in MRS broth at 37 °C for 24 h up to a cell concentration of 1 × 10^9^ cfu/mL. The growth ability was evaluated and reflected by absorbance (OD = 600 nm) (T6 UV Spectrophotometer, Beijing Puyan General Instrument Co., Ltd. Beijing, China) after incubation for 2, 4, 6, 8, 10, 12, 14, and 24 h [31].

#### 2.4.2. pH Determination of LAB Culture

Strains were grown in MRS broth at 37 °C for 24 h up to a cell concentration of 1 × 10^9^ cfu/mL. The pH was evaluated after each strain was incubated for 2, 4, 6, 8, 10, 12, 14, and 24 h. The pH was determined using a pH meter (Shanghai INESA Scientific Instrument Co., Ltd. Shanghai, China) [31].

### 2.5. Illumina High-Throughput Sequencing

The original sourdoughs, collected from XS and GS areas in the selected families, were firstly activated to mature dough (see Section 2.2). Next, the samples from the sourdough XS and GS after 0 h of fermentation were labeled as XS1 and GS1, respectively. The samples from the sourdough XS and GS after 4 h of fermentation were labeled as XS2 and GS2, respectively. For each sampling, three replicates were prepared. The genomic DNA of all samples was extracted and stored at −20 °C. Extracted DNA was used as the template, and the V4 region of bacterial 16S rRNA was amplified by PCR using the universal bacterial primers 515F and 806R with Illumina barcoded adapters. The two-step-PCR technology was conducted in which the first PCR step worked 25 cycles using untagged primers, the second PCR step worked 5 cycles with tagged primers, and the first step products were used as a template. The PCR amplification and DNA extraction were carried out according to the methods described by Francesca et al. [32]. The whole sequencing process was conducted using an Illumina MiSeq platform (Novogene Co., Ltd. Beijing, China).

The operational taxonomic unit (OTU) had an identity threshold of 97% with UPARSE software. Sample annotation analysis of representative sequences of OTUs was carried out with the highest frequency of occurrence using the Mothur method and SSUrRNA database (set threshold 0.8~1.0). Shannon index and Chao1 estimator values were calculated in RDP at 97% sequence similarity (http://pyro.cme.msu.edu/, 26 December 2019). Alpha diversity was evaluated by rarefaction curves, Good’s coverage, Simpson and Shannon diversity indices, and Chao1 richness. Principal coordinate analysis was used to visualize distance matrices and evaluate the global differences between samples [33,34,35,36].

### 2.6. Making Procedure of Steamed Bread Using LAB Sourdoughs

*Lactiplantibacillus plantarum* 23 and 2-2, *Pediococcus pentosaceus* 23, 2-7, and *Companilactobacillus crustorum* 22 were grown in MRS broth at 37 °C. Their cells were collected by centrifugation (4 °C, 4192× *g*, 10 min) and washed twice with sterile water. The cell concentrations of these tested strains were finally adjusted to 1 × 10^9^ cfu/mL prior to use.

The LAB strains-based sourdoughs were prepared by mixing 150 g of wheat flour, 1.5 g of dry yeast, 70 mL of water, and 5 mL of selected bacteria suspensions. The mixture was stirred in a spiral mixer (JHMZ-200, Beijing Oriental Fude Technology Development Co., Ltd. Beijing, China) for 10 min to smooth the surface of the dough. Next, the dough was divided into 50 g/piece and was placed in an incubator for fermentation at 37 °C with 80% relative humidity for 4 h [22]. The proofed dough was steamed for 20 min. Each dough treatment was processed in triplicate.

### 2.7. Textural Analysis of Steamed Bread

Textural analysis was performed according to the method described by Liu et al. [37]. A texture analyzer (TVT 6700, Perten, Sweden) was equipped with a P36 probe, and steamed bread was sliced vertically to obtain the middle pieces (25 mm thickness). The texture profile analysis (TPA) test profile gave the parameters of hardness, adhesiveness, flexibility, and chewiness. The test parameters were as follows: pre-test speed was 2.0 mm/s, test speed was 1.0 mm/s, post-test speed was 1.0 mm/s, and trigger force was 5 g. Compression time (the time of compressing the crumb center at 50% of the previous height) was 5 s [37]. Average values were obtained after triplicate measurements.

### 2.8. Image Analysis of Steamed Bread

Bread was cut into two halves vertically. The cut side of one of the halves was placed over the glass of a scanner. The scanned image was analyzed using the software ImageJ(National Institutes of Health, Maryland, USA). Average values were obtained after triplicate measurements. Image analysis was performed according to the method described by Semin et al. [38].

### 2.9. Volatile Compounds of Sourdough by GC-MS Analysis

Volatile compounds of different sourdoughs were determined by solid-phase microextraction in combination with the gas chromatography-mass spectrometry method (SPME-GC-MS). Sourdoughs were fermented at 37 °C and 80% relative humidity for 4 h. For each SPME analysis, 10 g of sourdough sample was added to a 50 mL vial, and the SPME needle was introduced through the vial septum. The vial was then immersed in a water bath at 60 °C, and the SPME fiber (2 cm–50/30 mm DVD/Carboxen/PDMS Stable Flex Supelco, Bellefonte, PA, USA) was exposed to the headspace for 60 min. When the extraction process was completed, the fiber was inserted into the injector port (set at 280 °C) of the GC (QP2010, Shimadzu, Japan) for thermal desorption of volatiles for 5 min in splitless mode. A Supelcowax-10 column (60 m, 0.32 mm i.d., 0.25 μm film thickness) was used. The GC temperature program was set as follows: 35 °C for 5 min, increased by 5 °C/min to 50 °C (held for 5 min), increased by 5.5 °C/min to 230 °C (held for 5 min). The total run time was 51.73 min. The carrier gas was helium with a flow rate of 2 mL/min. The interface temperature was 230 °C. Mass spectra were recorded by electronic impact at 70 eV, in the mass range of 33–200 m/z. The identification of volatile compounds was performed by comparison of the mass spectral data obtained with those of standard compounds and those in NIST11 and Wiley libraries [39].

### 2.10. Statistical Analysis

Graphical analyses and statistical analyses (ANOVA and Tukey test) were performed using Origin statistical software. Analyses were carried out in triplicate with three biological replicates for each condition. Significant differences were assessed at an error probability of 5% (*p* < 0.05).

### 2.11. Flowchart for the Present Study

We created a flowchart to explain the present study (Figure 1).

## 3. Results

### 3.1. Texture of Steamed Bread Made with Different Traditional Sourdoughs

Texture is one of the most important indicators for evaluating steamed bread quality. Figure 2 shows the texture properties of steamed bread, which include hardness, flexibility, adhesiveness, and chewiness. There was a significant change in the texture properties of steamed bread made with traditional sourdoughs compared to wheat steamed bread. The XSB and GSB reduced the hardness, flexibility, adhesiveness, and chewiness compared with WSB. There was no significant change in the hardness and adhesiveness of steamed bread between XSB and GSB. In comparison, XSB had the lowest flexibility and chewiness value.

From the results above, we can see that there was a significant change in the texture properties of steamed bread made with traditional sourdoughs when compared to wheat steamed bread. Based on LAB strains isolated from XS and GS, we made steamed bread made using sourdoughs of different LAB to compare with traditional sourdough. Figure 2 shows the texture properties of steamed bread, which include hardness, flexibility, adhesiveness, and chewiness. There was a tiny change in flexibility, adhesiveness, and chewiness of steamed bread made with *Lactiplantibacillus Plantarum* 20 from XS compared to *Lactiplantibacillus plantarum* 2-2 from GS, but there was no change in terms of hardness. There was a significant change in hardness, adhesiveness, and chewiness of steamed bread made with *Pediococcus pentosaceus* 23 from XS compared to *Pediococcus pentosaceus* 2-7 from GS. The steamed bread made with strains 2-2 and 2-7, which was isolated in GS, exhibited higher flexibility and adhesiveness compared with the steamed bread made with strains 20, 22, and 23, isolated in XS. The unexpected outcome was that the texture difference between two strains from different regions was significant, and the correlation between strain-specific and texture properties is interesting.

### 3.2. Image Analysis of Steamed Bread Made with Different Traditional Sourdoughs

Macro-structure of the steamed bread was obtained by Image analysis. The pore diameter range was within the macro-scale pore diameter range, which was obtained by the software ImageJ (National Institutes of Health, Bethesda, MD, USA).

The pores of wheat steamed bread are dense and uniform, without obvious irregular large bubbles, while the pores and distribution of steamed bread made with traditional sourdoughs are uneven, and there are irregular large and sparse bubbles (Figure 3). The data shown in Table 2 can more intuitively reflect the difference in the internal texture of different steamed breads. The area fraction of the pore surface of WSB is 9.78%, the average diameter of the pore is 74.25 µm, and the density of the pore is 102 PPI. The area fraction of the pore surface of XSB is 11.22%, the average diameter of the pore is 79.06 µm, and the density of the pore is 103 PPI. The area fraction of the pore surface of GSB is 12.48%, the average diameter of the pore is 90.73 µm, and the density of the pore is 105 PPI. Overall, these results indicate that the area fraction of the pore surface, the average diameter of the pore, and the density of the pore of steamed bread made with traditional sourdoughs are significantly higher than the others.

As shown in Figure 3, the pores and distribution of steamed bread made with LAB doughs are uneven, and there are irregular large and sparse bubbles. Table 2 shows the area fraction of the pore surface of 23 is 10.72%, the average diameter of the pore is 53.00 µm, and the density of the pore is 113 PPI. The area fraction of the pore surface of 2-7 is 10.57%, the average diameter of the pore is 50.00 µm, and the density of the pore is 105 PPI. Strains 23 and 2-7 both belong to *Pediococcus pentosaceus* and have similar internal texture structures. However, the area fraction of the pore surface of 20 is 13.53%, the average diameter of the pore is 65.00 µm, and the density of the pore is 98 PPI. The area fraction of the pore surface of 2-2 is 9.64%, the average diameter of the pore is 35.00 µm, and the density of the pore is 121 PPI. It can be seen that the steamed bread made with *Lactiplantibacillus plantarum* 2-2 has a smaller area fraction of the pore surface and larger density.

### 3.3. Volatile Compounds of Different Traditional Sourdoughs

Simple statistical analysis was used to determine volatile compounds of different traditional sourdoughs (Figure 4a). A total of 44 volatile compounds were identified in WSB, containing 12 esters, 1 acid, 13 alcohols, 1 ether, 1 ketone, 14 alkanes, and 4 aromatic heterocycles. A total of 50 volatile compounds were identified in XSB, containing 12 esters, 4 acids, 14 alcohols, 3 aldehydes, 2 ketones, 10 alkanes, and 5 aromatic heterocycles. A total of 51 volatile compounds were identified in GSB, containing 13 esters, 2 acids, 16 alcohols, 4 aldehydes, 3 ketones, 6 alkanes, and 3 aromatic heterocycles.

As shown in Figure 4b, the most striking observation to emerge from the data comparison was that the richness of volatile matter in traditional sourdough is higher than in yeast dough. Further analysis showed that the similarity of volatile compounds in two types of sourdough is higher than that of yeast dough. Aldehyde was only detected in XSB and GSB, and WSB contains more kinds of alkanes. There are slight differences between XSB and GSB. Strong evidence shows that traditional sourdough has better flavor, but the volatile compounds of different sourdoughs exhibit great differences.

Figure 4a presents volatile compounds of steamed bread made of different LAB compared to traditional sourdoughs. A total of 42 kinds of volatile compounds were identified in strain 20 dough, containing 10 esters, 4 acids, 10 alcohols, 1 ether, 2 aldehydes, 12 alkanes, and 3 aromatic heterocycles. A total of 41 volatile compounds were identified in strain 22 dough, containing 9 esters, 5 acids, 10 alcohols, 1 ketone, 15 alkanes, and 1 aromatic heterocycle. A total of 47 volatile compounds were identified in strain 23 dough, containing 9 esters, 3 acids, 13 alcohols, 1 aldehyde, 1 ether, 1 ketone, 17 alkanes, and 2 aromatic heterocycles. A total of 46 volatile compounds were identified in strain 27 dough, containing 9 esters, 4 acids, 15 alcohols, 1 aldehyde, 16 alkanes, and 1 aromatic heterocycle.

The volatile profile of the four LAB doughs studied was analyzed (Figure 4b). The most abundant volatile compounds found in all varieties of LAB doughs were acids and alkanes, while volatile compounds were balanced in traditional sourdough. In general, the LAB doughs induced a greater content of the single volatile compounds identified compared to the traditional sourdough. Although 20 and 2-2 are *Lactiplantibacillus plantarum* and 23 and 2-7 are *Pediococcus pentosaceus*, the volatile components of LAB dough were different. 3-hydroxybutanal and cis-3-nonene-1-ol were only detected in 20, 23, and XSB, methoxybenzoxime, acetaldehyde diethanol, and pentanol were only detected in 2-2, 2-7, and GSB. This indicated that 20, 23, and XSB had special volatile substances along with 2-2, 2-7, and GSB. Together these results provide important insights into the difference of volatile compounds, which is related to the microbial community composition of sourdough.

### 3.4. Identification of LAB Strains Isolated from Sourdough

A total of 35 microorganism isolates recovered from traditional sourdough samples were microscopically identified as LAB (Table 3 and Table 4). Comparison results of the homologous strains of the isolated sourdough LAB are shown in Figure 5.

It is apparent from this table that the LAB isolates recovered from traditional sourdoughs belong to two genera, four species or subspecies, and the homology of the species is as high as 100%. Sixteen LAB strains were identified from XS, chiefly *Lactiplantibacillus plantarum*, *Pediococcus pentosaceus*, and *Companilactobacillus crustorum*. Nineteen LAB strains were identified from XS, chiefly *Lactiplantibacillus plantarum* and *Pediococcus pentosaceus*. *Lactiplantibacillus plantarum* was the dominant strain in XS which accounted for 81.25% of the LAB isolated. *Lactiplantibacillus plantarum* was the dominant strain in GS which accounted for 63.16% of the LAB isolated, and *Pediococcus pentosaceus* was the subdominant strain in GS, which accounted for 36.84% (Figure 5).

### 3.5. The Optimal LAB Strains for Growth and pH

Analysis of the growth performance of the isolated sourdough LAB in GS at different times (Appendix A) revealed that the 2-2 strain had the best growth ability, and the 21 strain also had good growth ability. Regarding the ability of selected LAB to grow under acidic environments, strains 22, 27 had the most rapid decrease in pH (Appendix A). Therefore, strain 22 is a fast-growing and acid-producing strain in GS. Analysis of the growth performance of the isolated sourdough LAB in XS at different times (Appendix A) revealed that strains 18 and 21 displayed great growth ability. Concerning the ability of selected LAB to grow under acidic environments, strains 20 and 23 showed the most rapid decrease in pH (Appendix A). As shown in Appendix A, *Lactiplantibacillus plantarum* 22, *Lactiplantibacillus plantarum* 21, and *Pediococcus pentosaceus* 27 are dominant strains in the sourdough GS, while strains *Lactiplantibacillus plantarum* 3, *Lactiplantibacillus plantarum* 18, *Lactiplantibacillus plantarum* 20, and *Pediococcus pentosaceus* 23 dominate the sourdough XS. Clearly, *Lactiplantibacillus plantarum* 23 and 22, as observed regarding their growth capacity and pH determination experiments, should be selected as the dominant strain of the sourdoughs, and *Pediococcus pentosaceus* 23, 27, is another dominant species responsible for the dough acidification.

### 3.6. Microbial Community Diversity Analysis in Sourdough

Microbial communities of these two types of traditional sourdoughs were investigated by Illumina high-throughput sequencing technology. For each sourdough, the microbial community composition at the genus level is shown in Figure 6a. There was a significant difference in the number of bacterial OTUs between the two regions’ sourdoughs. *Lactobacillus* was the most abundant genus in group XS (more than 75%), while *Pediococcus* was the minority genus (1%). *Pediococcus* was the most abundant genus in the group GS (more than 45%), while *Lactobacillus* was the minority genus (1%). The single most striking observation to emerge from the data comparison was that LAB is the core bacterial species in XS and GS, but predominant strains at the genus level in sourdough vary by region.

Illumina high-throughput sequencing can comprehensively reveal the species composition of bacteria at the species level (Figure 6b). *Lactobacillus* was composed of *Lactiplantibacillus plantarum*, *Companilactobacillus crustorum*, *Lactiplantibacillus pentosus*, and *Levilactobacillus brevis* at class level, while *Pediococcus* was all composed of *Pediococcus pentosaceus*. Based on our data, *Lactiplantibacillus plantarum* is the core bacterial species in XS (larger than 65%), and *Pediococcus pentosaceus* was relatively abundant in GS (larger than 39%), but *Lactiplantibacillus plantarum* was very limited in GS. It can be seen from the data in Table 5 that fermentation time will affect the microbial community diversity of sourdough. With successive increases in the fermentation time, the relative abundance of *Lactiplantibacillus plantarum* in XS increased further (from 67.65% to 72.23%). However, the relative abundance of *Companilactobacillus crustorum*, *Lactiplantibacillus pentosus*, and *Levilactobacillus brevis* in XS showed a decreasing trend. Interestingly, the relative abundance of *Pediococcus pentosaceus* in GS reduced (from 45.66% to 39.80%). However, the relative abundance of *Lactobacillus* in GS had no significant change. Overall, these results indicate that *Lactiplantibacillus plantarum* was the dominant bacteria species in XS, *Pediococcus pentosaceus* is the dominant bacteria species in GS, and *Lactiplantibacillus plantarum* and *Pediococcus pentosaceus* were observed in all sample groups. This result was consistent with the data obtained by LAB isolation methods.

### 3.7. Alpha Diversity and Beta Diversity in Sourdough

As shown in Figure 7, alpha diversity indices of OTUs from samples were expressed with Chao, Shannon, and Simpson and observed species parameters. All alpha diversity indices were lower in XS than in GS, which indicates that the species diversity in GS is more abundant. The Chao1 richness estimator for XS2 was 237.59, higher than the XS1 (235.805). The Simpson index for XS2 was 0.603, compared to a value of 0.605 for XS1. The Shannon index for XS2 was 2.723, compared to a value of 2.714 for XS1. It was obvious that XS1 was richer than XS2 in species. The Chao1 richness estimator for GS1 was 262.563, higher than the GS2 (261.968). The Simpson index for GS2 was 0.785, compared to a value of 0.737 for GS1. The Shannon index for GS2 was 3.504, compared to a value of 3.232 for GS1. It was obvious that GS2 was richer than GS1 in species. The species diversity of XS decreases, while the species diversity of GS increases after 4 h of fermentation.

Principal component analysis (PCA) showed that in samples from the same region, XS1 and XS2 were similar in microbial community composition and clustered together, which was similar to samples of GS1 and GS2 (Figure 8a). On average, there was a great distance between XS1 and XS2, and GS1 and GS2, and there was a significant difference in microbial composition between the two kinds of sourdoughs. The correlation coefficient value was 0.023 between XS1 and XS2, and the correlation coefficient value was 0.037 between GS1 and GS2 (Figure 8b). The results indicate that XS1 and XS2 were more similar in microbial community composition than GS1 and GS2. The correlation coefficient value was 0.232 between XS1 and GS2, which was the maximum value. There was a significant difference in microbial composition between XS1 and GS2. The results were consistent with the PCA analysis.

## 4. Discussion

Chinese steamed bread is one of China’s important staple foods, and prior studies have noted the importance of the quality of steamed bread [40,41]. In general, the quality of steamed bread has been evaluated using the industry standard SB/T10139-93 of the ministry of commerce of China or the national standard GB/T17320-1998. One major drawback of these standards is their focus on sensory evaluation [42,43]. However, this method does involve potential measurement errors from the subjective influence of testers. The quality of the study would have been more rigorous and accurate if it had included texture analysis, pore structure analysis, and analysis of volatile compounds by GC-MS [38,44,45].

The quality of steamed bread made with traditional sourdoughs was superior to wheat steamed bread. In comparison, XSB also showed a significant difference compared with GSB. Hardness was recognized as a marker to reflect the flavor and freshness of bread [46]. Gupta et al. presented that both citric acid and malic acid could significantly reduce the hardness of bread [47]. *Lactiplantibacillus plantarum* DM616 decreased the hardness of steamed bread after 16 h of fermentation compared with the control [31]. Previous studies have demonstrated that LAB fermentation increases the acidity of the oatmeal sourdough, and acidity causes the adhesiveness of sourdough to decrease by directly affecting the properties of the protein [48]. Studies revealed significantly softer crumb and lower chewiness and resilience values in bread containing 20% sourdough compared with standard bread [49]. Hence, it could conceivably be hypothesized that the change in the texture of the steamed bread is related to the acidity of its sourdough. The significant role of LAB has been confirmed to explain the observed differentiation in pore structure in sourdough bread [50]. The fermentation and metabolism of LAB contribute to the stomata of the bread, affecting the fine and regular texture of sourdough [7]. The presence of LAB strains enhances the activity of the yeasts to increase the dough volume and CO_2_ production [51]. A study implies that the *Pediococcus pentosaceus* strain, which possesses specific proteolytic activities, could weaken the gluten network. The weakening of the gluten network structure would easily break the film of stomata, thereby forming a sparse atmosphere pore [52]. The literature showed differences in the volatile composition of the bread of different mixed starter cultures of sourdough [39]. This study showed that steamed bread with LAB starter contained the highest content of volatile compounds [53]. All bread produced by the traditional sourdough method had a different aroma profile, with more volatiles identified, including alcohols and ester compounds. This finding was also reported by Stavros et al. [54]. These results further support the hypothesis that the difference in the quality of steamed bread is due to variations in microbial populations.

Combining Illumina high-throughput sequencing and conventional culturable methods could obtain more comprehensive and accurate microbiome information [55]. These studies investigated microbial communities in traditional sourdoughs using conventional culturable methods and next-generation sequencing [56,57,58]. Hence, the methods were used to explore the relationship between bread quality and microbial communities of sourdough. The microbial communities were diverse in the two types of sourdough in this work. The identified results showed that the pure isolates were *Lactiplantibacillus plantarum*, *Pediococcus pentosaceus*, and *Companilactobacillus crustorum,* according to the culturable method. Among them, *Lactiplantibacillus plantarum* was the dominant strain in traditional Chinese sourdough, and the results agreed with the findings of Zhang et al. [59]. We found that *Pediococcus pentosaceus* was the second most dominant species isolated in GS after *Lactiplantibacillus plantarum*. *Companilactobacillus crustorum* was also an isolated strain in XS. This strain was first found in sourdough in Belgium, and it has been frequently found in sourdough in various places [60]. The types and quantities of isolates in the two sourdoughs are dissimilar. At this point, the communities of bacteria identified from high-throughput sequencing might provide more detail on the dynamics of the microbial communities from two kinds of sourdoughs. The results showed that *Pediococcus pentosaceus* was the core strain in GS, and *Lactiplantibacillus plantarum* was the dominant bacteria in XS. However, according to our observations (see Section 3.4), which showed that *Lactiplantibacillus plantarum* was the dominant strain in XS and GS when using the culturable method, the findings of high-throughput sequencing do not support the results using the culturable method. A possible explanation for this is that *Pediococcus pentosaceus* is particular with some nutrients and needs to be incubated in specific media [61]. Another possible explanation for this is that *Pediococcus pentosaceus* had slower growth compared to the other LAB isolates [62]. The above explanations should be the main reasons for the difference in the results of conventional culturable methods.

Combining the two methods, we determined that the core bacteria was *Pediococcus pentosaceus* in GS, and the core bacteria was *Lactiplantibacillus plantarum* in XS. A study revealed the predominant microbes in five traditional Chinese sourdoughs were *Lactobacillus*, *Pediococcus*, and *Wickerhamomyces* [63]. The results also showed that the traditional Chinese sourdoughs were significantly different from different regions, and the dominant genera were *Lactobacillus*, *Pediococcus*, and *Leuconostoc*, and this also is in accordance with our earlier observations [45]. Another study proved that the microbial community’s diversity in sourdoughs can vary according to geographic location [64]. According to these results and the literature, we can infer that the differences in the dominant microbiota are related to the differences in the collection regions.

As the fermentation time went by, the microbial community from two kinds of sourdoughs also exhibited dynamic change. After 4 h of fermentation, the relative content of the dominant strain in XS (*Lactiplantibacillus plantarum*) increased, and the relative content of the dominant strain in GS (*Pediococcus pentosaceus*) decreased. This discrepancy could be attributed to the difference in dominant strains in sourdough. Antagonism between *Lactiplantibacillus plantarum* and *Pediococcus pentosaceus* has been observed [65]. On the other hand, *Lactiplantibacillus plantarum* has a faster acid-producing ability, which lowers the pH value in sourdough. *Lactiplantibacillus plantarum* can utilize more nutrients in its habitat that reduce the growth rates of other bacteria [66,67,68]. This finding, while preliminary, suggests that the dominant bacteria significantly affect the microbial community in sourdough.

The alpha diversity was investigated in sourdoughs divided according to the different kinds (XS and GS) and fermentation time. Unexpectedly, there was only a slight variation in the diversity between XS and GS with different fermentation times, but the species diversity in GS is more abundant compared to XS. The finding showed that bakers’ practices and status, as considered here, did not appear to be significantly related to LAB diversity and sourdough features [69]. There is no relationship between the time of chief sourdough storage or the backslopping frequency and the microbial community, although a longer period between backslopping may generate more stressful conditions for sourdough microbiota (e.g., low pH, lack of nutrients) [58]. The eventually established microbial community commonly reflects the media resources (carbohydrates, amino acids, vitamins) and environmental conditions (temperature, pH, redox potential) [70]. In terms of media resources and environmental conditions, sourdough had regional specificity. We suggested that geographical sites might impact the microbial community more than fermentation time. The beta diversity was represented by PCA and a heat map. The results of the PCA and heat map further support the conjecture.

A comparison in the quality of dough fermented with dominant strains and traditional sourdoughs was made. There were only slight differences between the same LAB species, but their quality was strain-specific. This study confirms that flexibility is associated with the addition of acids in doughs, and the greatest contributing factor was decreasing phase angle values [71]. One study demonstrated an increase in both the softness and elasticity of gluten in the presence of acid [72]. In accordance with the present results, previous studies have demonstrated that depending on the rate of acid production, the bread quality will also vary [31,73]. Thus, a possible explanation for these results is the capability of acid production from LAB, which varies. Hadaegh et al. found that the pH of sourdoughs with the addition of LAB cultures decreased compared to the dough without LAB, and sourdoughs with various LAB showed different ranges of pH and total titratable acidity (TTA) [73]. As mentioned in the literature, a possible explanation for the better resistance to spoilage of the sourdough bread made with immobilized *Pediococcus pentosaceus* SP2 sourdough is its higher TTA and organic acid content [74]. The lupine sourdoughs made with *Pediococcus* strains resulted in the highest specific volume and crumb porosity compared to lupine flour bread made with LAB [75]. These findings, while preliminary, suggest that the main reason for a better quality of steamed bread is the strong acid-producing ability of the dominant strain. The capacity of acid production by LAB can vary according to different microbial community’s diversity in sourdoughs.

Sourdoughs are very complex mixed microbial ecosystems. An interesting finding is that the flavor maps are distributed more equally when heterofermentative *Lactobacillus* and homofermentative *Lactobacillus* are co-fermented [13]. This also is in accordance with our observations (see Section 3.3), which showed that the volatile substance distributions in XSB or GSB were more balanceable than in WSB. According to the literature, the higher the complexity of volatile compounds, the better the aroma of bread [76,77]. Previous research has confirmed that hetero- and homofermentative strains have different effects on sourdough flavor [78,79]. It has been suggested that bread samples could form different aroma profiles due to the clustering based on the differential composition of LAB in their respective starter cultures [80]. Xi’s finding showed that while some important aroma compounds found in both CSB were the same, they were significantly different in their concentrations [28]. These studies match those observed in our results (see Section 3.3). Therefore, we conclude that differences in the microbiota and the dominant strains lead to differences in the quality of steamed bread from region to region.

## 5. Conclusions

The main goal of the current study was to detect quality differences in sourdough produced in different regions. The microbial communities of LAB from sourdough from different locations in Northwestern China were comprehensively revealed through a combination of high-throughput sequencing and culture-dependent methods. The effects of LAB on the volatile flavor compounds of sourdough were preliminarily analyzed, and the dominant strains were selected as starter cultures for steamed bread production and compared to those of traditional sourdoughs. This study identified that sourdough-based steamed bread is superior in terms of hardness, adhesiveness, flexibility, chewiness, and content of volatile substances compared to yeast-based steamed bread. *Lactiplantibacillus plantarum* and *Pediococcus pentosaceus* were the dominant species found in samples of Gansu sourdough, accounting for 42.11% and 36.84% of the total, respectively. *Lactiplantibacillus plantarum* was the dominant species, accounting for 75.00% of species seen in samples of Xinjiang sourdough. It is clear that the species diversity from Gansu sourdough is higher. Compared to the acid production ability of different strains during the fermentation process of sourdough, *Lactiplantibacillus plantarum* strains were regarded as the strongest acid producers. The volatile profiles of sourdoughs were similar, but differences in individual volatile compounds between the sourdoughs of the Gansu and Xinjiang regions were detected. These results point out that differences in the microbiota and the dominant strains may lead to differences in the quality of steamed bread from region to region.

In summary, this work systematically analyzed the microbiota structure of sourdough in the Xinjiang and Gansu regions. It provides a theoretical basis and technical support for the use of dominant LAB blend cultures in the processing of traditional steamed bread. In a word, our work provides new insights into improving the quality of traditional steamed bread by adjusting the microorganisms in sourdough. This work should be of great significance for the large-scale processing of typical steamed bread.

## Figures and Tables

**Figure 1 foods-11-01908-f001:**
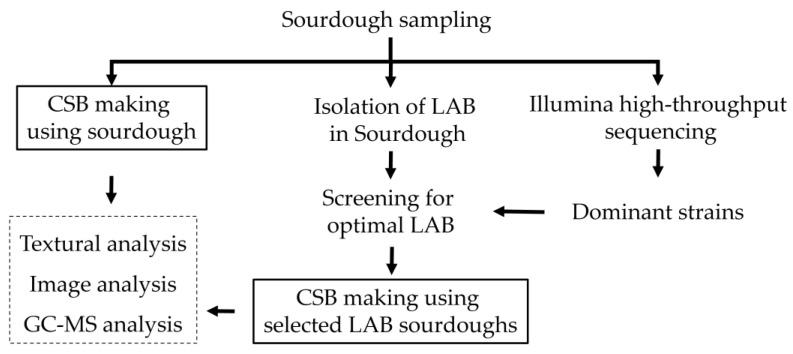
Flowchart for the present study.

**Figure 2 foods-11-01908-f002:**
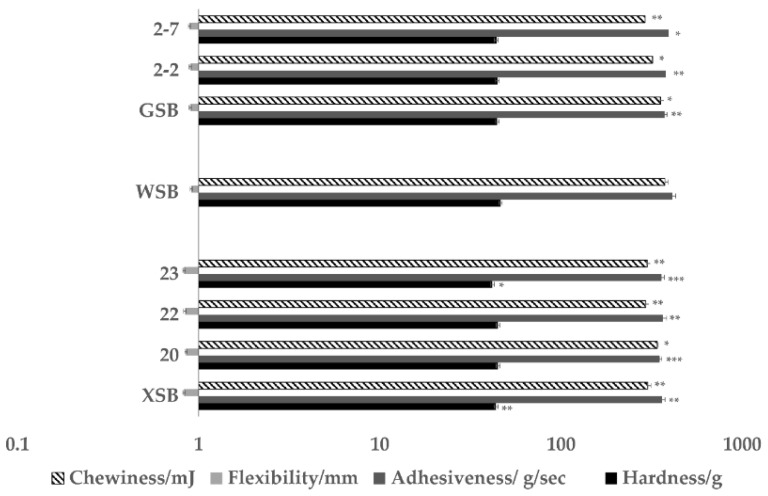
Survey results of texture characteristics of different sourdough steamed bread. Note: a *, **, and *** indicate significance at *p* = 0.05, 0.01, and 0.001, respectively.

**Figure 3 foods-11-01908-f003:**
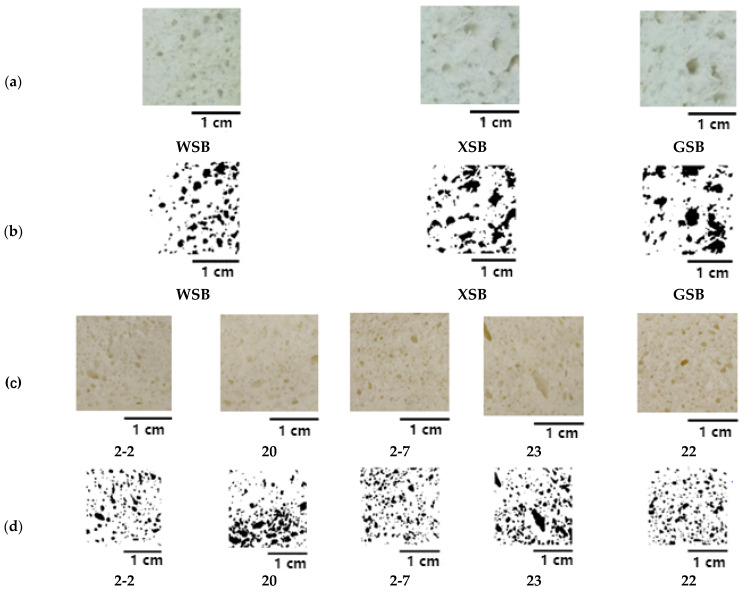
Image of stomatal structure of steamed bread core of different sourdough steamed bread ((**a**): 2D plane scan; (**b**): analytical graph after gray treatment; (**c**): 2D plane scan; (**d**): analytical graph after gray treatment).

**Figure 4 foods-11-01908-f004:**
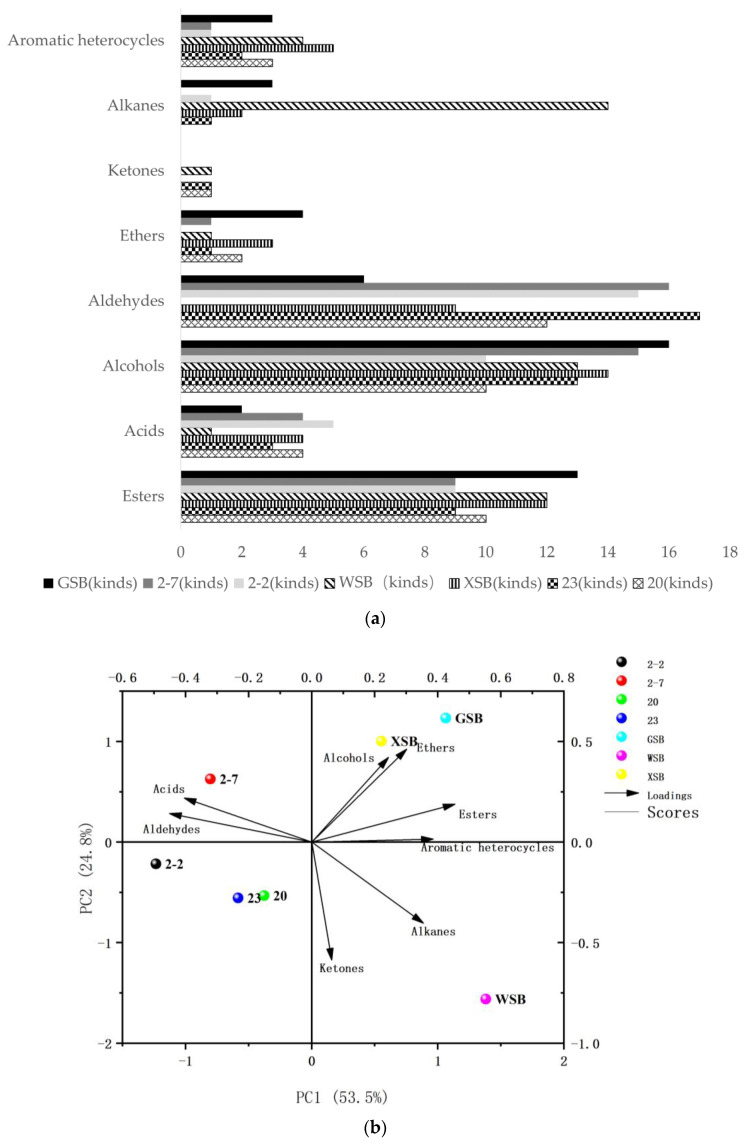
Statistical results of volatile compounds of different traditional sourdoughs ((**a**): bar chart; (**b**): principal correspondence analysis (PCA)).

**Figure 5 foods-11-01908-f005:**
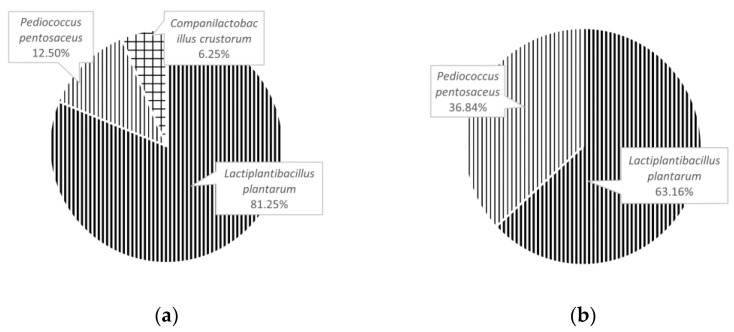
The percentage of different species or subspecies in the total strains isolated from XS and GS ((**a**): XS, (**b**): GS).

**Figure 6 foods-11-01908-f006:**
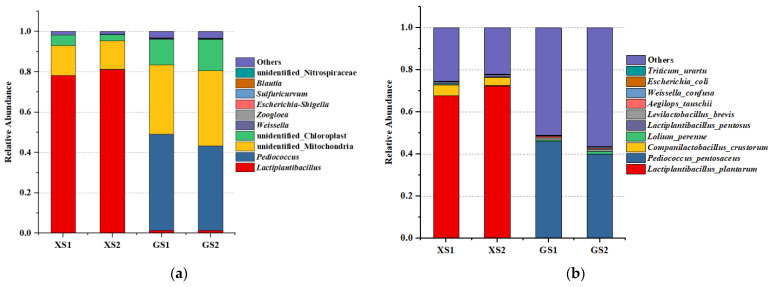
Relative abundance of microbial communities in different sourdoughs ((**a**): at the genus level, (**b**): at the species level).

**Figure 7 foods-11-01908-f007:**
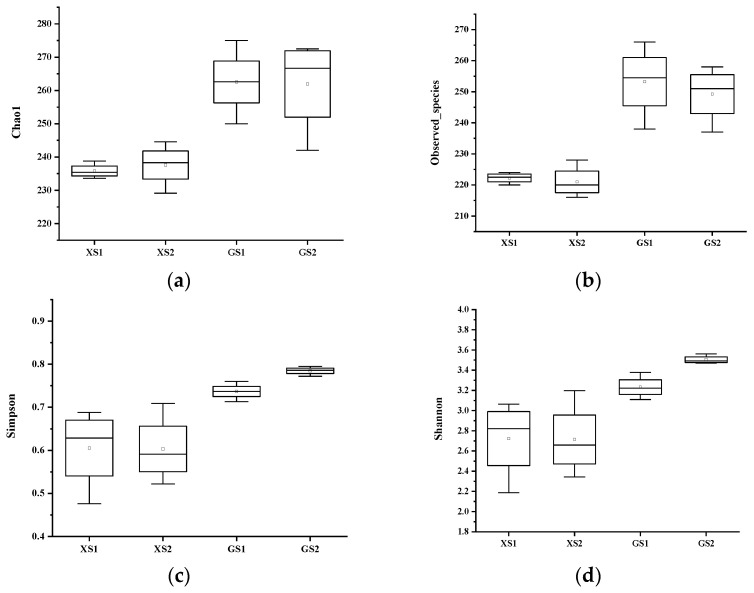
Analysis of bacterial abundance and diversity index of two sourdough fermentation periods. ((**a**) represents the changing trend of the Chao1 index; (**b**) represents the observed species change trend; (**c**) represents the Simpson index variation trend; (**d**) represents the variation trend of the Shannon index).

**Figure 8 foods-11-01908-f008:**
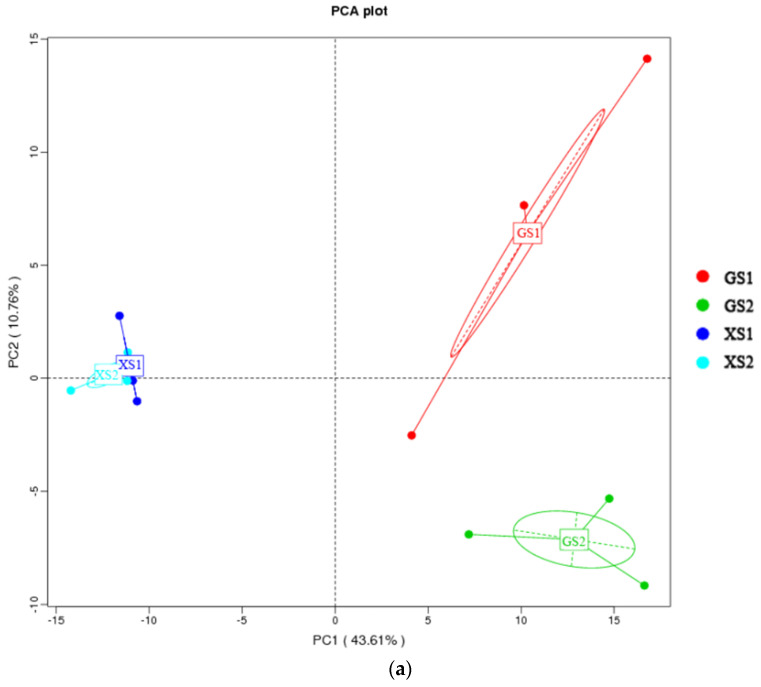
Beta diversity of the sourdough ((**a**): principal correspondence analysis (PCA); (**b**): heat map).

**Table 1 foods-11-01908-t001:** Recipe for fermented dough.

	Wheat Flour/g	Sterile Water/mL	Yeast/g	Sourdough/g
WSB	150	75	1.5	/
GSB	150	70	/	5
XSB	150	70	/	5

Note: The 3 types of bread are as follows, yeast bread (WSB), Gansu sourdough bread (GSB), and Xinjiang sourdough bread (XSB), “/“ indicate the number of gram is zero.

**Table 2 foods-11-01908-t002:** Image analysis of steamed bread.

	Area Fraction of the Pore Surface/%	Average Diameter of the Pore/µm	Density of the Pore/PPI
WSB	9.78 ± 0.56	74.25 ± 0.08	102.00 ± 48.71
XSB	11.22 ± 0.73	79.06 ± 0.08	103.00 ± 51.25
GSB	12.48 ± 0.41	90.73 ± 0.09	105.00 ± 45.95
20	13.53 ± 2.11	65.00 ± 0.08	98 ± 49.21
2-2	9.64 ± 3.69	35.00 ± 0.04	121 ± 34.56
23	10.72 ± 2.99	53.00 ± 0.05	113 ± 39.35
2-7	10.57 ± 1.68	50.00 ± 0.04	105 ± 42.50
22	10.22 ± 0.96	46.00 ± 0.04	117 ± 40.40

**Table 3 foods-11-01908-t003:** Homology analysis of LAB strains from Xinjiang sourdough.

Number	Homologous Strain	Sequence Integrity	Similarity
2	*Lactiplantibacillus plantarum* ATCC 8014	complete genome	100%
3	*Lactiplantibacillus plantarum* ATCC 8014	complete genome	100%
4	*Lactiplantibacillus plantarum* ATCC 8014	complete genome	100%
5	*Lactiplantibacillus plantarum* ATCC 8014	complete genome	100%
9	*Lactiplantibacillus plantarum* ATCC 8014	complete genome	100%
12	*Lactiplantibacillus plantarum* ATCC 8014	complete genome	100%
13	*Lactiplantibacillus plantarum* ATCC 8014	complete genome	100%
15	*Lactiplantibacillus plantarum* ATCC 8014	complete genome	100%
16	*Lactiplantibacillus plantarum* ATCC 8014	complete genome	100%
17	*Lactiplantibacillus plantarum* ATCC 8014	complete genome	100%
18	*Lactiplantibacillus plantarum* ATCC 8014	complete genome	100%
19	*Lactiplantibacillus plantarum* ATCC 8014	complete genome	100%
20	*Lactiplantibacillus plantarum* TMW 1.1623	complete genome	100%
21	*Pediococcus pentosaceus* ATCC 25745	complete genome	100%
23	*Pediococcus pentosaceus* ATCC 25745	complete genome	100%
22	*Companilactobacillus crustorum* KCC-12	complete genome	100%

**Table 4 foods-11-01908-t004:** Homology analysis of LAB strains in Gansu sourdough.

Number	Homologous Strain	Sequence Integrity	Similarity
2-1	*Lactiplantibacillus plantarum* ATCC 8014	complete genome	100%
2-13	*Lactiplantibacillus plantarum* ATCC 8014	complete genome	100%
2-19	*Lactiplantibacillus plantarum* ATCC 8014	complete genome	100%
2-21	*Lactiplantibacillus plantarum* ATCC 8014	complete genome	100%
2-2	*Lactiplantibacillus plantarum* TMW 1.1623	complete genome	100%
2-5	*Lactiplantibacillus plantarum* TMW 1.1623	complete genome	100%
2-8	*Lactiplantibacillus plantarum* TMW 1.1623	complete genome	100%
2-11	*Lactiplantibacillus plantarum* TMW 1.1623	complete genome	100%
2-12	*Lactiplantibacillus plantarum* TMW 1.1623	complete genome	100%
2-15	*Lactiplantibacillus plantarum* TMW 1.1623	complete genome	100%
2-16	*Lactiplantibacillus plantarum* TMW 1.1623	complete genome	100%
2-17	*Lactiplantibacillus plantarum* TMW 1.1623	complete genome	100%
2-3	*Pediococcus pentosaceus* ATCC 25745	complete genome	100%
2-7	*Pediococcus pentosaceus* ATCC 25745	complete genome	100%
2-9	*Pediococcus pentosaceus* ATCC 25745	complete genome	100%
2-10	*Pediococcus pentosaceus* ATCC 25745	complete genome	100%
2-14	*Pediococcus pentosaceus* ATCC 25745	complete genome	100%
2-20	*Pediococcus pentosaceus* ATCC 25745	complete genome	100%
2-22	*Pediococcus pentosaceus* ATCC 25745	complete genome	100%

**Table 5 foods-11-01908-t005:** Relative abundance of different sourdough samples at the classification species level.

Strains	Relative Abundance ± SD (%)			
	XS1	XS2	GS1	GS2
*Lactiplantibacillus plantarum*	67.65 ± 0.08	72.23 ± 0.02	0.16 ± 0.00	0.24 ± 0.00
*Pediococcus pentosaceus*	0.12 ± 0.00	0.24 ± 0.00	45.66 ± 0.03	39.80 ± 0.03
*Companilactobacillus crustorum*	5.03 ± 0.02	4.00 ± 0.01	0.02 ± 0.00	0.02 ± 0.00
*Lactiplantibacillus pentosus*	1.11 ± 0.00	0.96 ± 0.00	0.02 ± 0.00	0.02 ± 0.00
*Levilactobacillus brevis*	0.03 ± 0.00	0.03 ± 0.00	0.97 ± 0.00	0.09 ± 0.00

## Data Availability

Data is contained within the article or Appendix A.

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
