# Peer review of "Relationship between Microbial Composition of Sourdough and Texture, Volatile Compounds of Chinese Steamed Bread"

_foods, 2022, doi:10.3390/foods11131908_

Round 1
Reviewer 1 Report
The work is interesting, proving further information on the microbial community of Chinese sourdough-based steamed bread.
Most of the comments have been highlighted within the manuscript to improve the work.
Other comments are:
Be specific throughout the manuscript that it is sourdough steamed bread you are referring to.
Improve on the abstract by introducing the topic shortly, the brief method and the findings before conclusion.
Consistency in the spelling of sourdough is essential.
Discussions in the result section should be moved to discussion section. Alternatively, Results and Discussion can be merged and properly sequentially arranged.

Author Response
Dear Editors and Reviewers,
Thank you very much for your letter with comments on our manuscript entitled “Relationship Between Microbial Composition of Sourdough and Texture, Volatile Compounds of Chinese Steamed Bread” (Foods). These comments are all valuable and helpful for revising our manuscript, and importantly for directing our future researches. We have revised our manuscript according to reviewer’s comments, and would like to re-submit it for your consideration. We have responded the comments raised by the reviewer’s points by points. The amendments are highlighted in red in the revised manuscript. The followings are main corrections in the manuscript and the responds to the reviewer’s comments.
Responds to the reviewer’s comments:
- Response to comment: (Title needs modification.
Suggestion:
Relationship between microbial composition of sourdough, texture and volatile compounds of Chinese steamed bread)
Response: Thank you for the title suggested. The precedent version of the title has been replaced, becoming “Relationship Between Microbial Composition of Sourdough and Texture, Volatile Compounds of Chinese Steamed Bread”.
- Response to comment: (This statement needs re-structuring to capture the conclusion in terms of what specifically gave the advantage in a non-traditional steamed bread)
Response: Thank you for the suggestion. We have modified the sentence according to the comment. Line 24 to 26, the statements of “The investigation offered a promising idea to improve traditional steamed bread's technological and functional properties.” were corrected as “This investigation offers a promising guidance on how to improve the quality of traditional steamed breads through adjusting the microorganisms in sourdough.”.
- Response to comment: (flour is more appropriate)
Response: We have changed it according to the suggestion(Line 30).
- Response to comment: (This sentence is hanging and unclear)
Response: Thank you for the suggestion. We have modified the sentence according to the comment. Line 55 to 56, the statements of “Leuconostoc citreum HO12 and Weissella koreensis HO20 as starter cultures in the making of whole wheat sourdough bread and production of sourdough breads with overall satisfactory quality [16].” were corrected as “Leuconostoc citreum HO12 and Weissella koreensis HO20 isolated from kimchi were assessed as starter cultures in the making of sourdough bread[16].”.
- Response to comment: (Be more explicit. Are you talking about Most studies in Asia or all continents?)
Response: Thank you for the suggestion. We have rewrote the introduction according to the comment. We are talking about most studies in China. Please see Line 72 to 73.
- Response to comment: (good quality of what?)
Response: We have modified the sentence according to the comment. Line 65 to 68, the statements of “The northwestern China wheat region, including most of Gansu, Xinjiang, and Ningxia, is characterized by high yield potential and good quality[20].” were corrected as “Local people in Northwestern China including Gansu, Xinjiang, and Ningxia, Qinghai and Shaanxi areas where high-quality and high-yielding wheat is produced, almost enjoy sourdough-based steamed buns every day [26].”.
7. Response to comment: (Sentence is not clear. kindly re-write.)
Response: We have rewrote the section according to the reviewer’s comment. Please see Line 78 to 82.
- Response to comment: (Sentences appear as conclusions. The insertion of "could" is more appropriate here.
Any conclusion should come at conclusion section out under ABSTRACT.)
Response: We really thank for this comment. We have rewrote the section according to the reviewer’s comment. We deleted this sentence.
- Response to comment: (Fragment the sentences to make better meaning.
How many sourdoughs were collected from each family?)
Response: We are grateful for the suggestions. The section 2.1 had be separated from this part and mentioned in a new section as follows: “
2.1 Sourdough sampling
The sourdoughs analyzed in this study were collected from different families located in the provinces of Xinjiang (XS) and Gansu (GS), which produced sourdough bread without any added salt. The selected families have been making Chinese steamed bread (CSB) with sourdough every day for more than 10 years. Three sour-doughs were collected from each family. All samples were kept at low temperature (4 °C) during transportation and storage.
2.2 Chinese steamed breadmaking using sourdough
To evaluate the quality of the products made using sourdough from different provinces, yeast CSB was added as blank control. The recipes of yeast CSB (WSB), Gansu sourdough bread (GSB), and Xinjiang sourdough bread (XSB) are shown in Table 1.
The sourdough, basically classified as type I sourdough. Firstly, the samples of sourdough were activated with sourdough and sterile water in the mass ratio of 2:3 and fermented at around 25 ℃ for 9 to 24 h to obtain a mature sourdough. Secondly, dough was prepared as described by Xi et al. (2020) with some modifications (see recipe in Table 1) [29]. Dough was prepared in a spiral mixer (JHMZ-200, Beijing Oriental Fude Technology Development Co., Ltd. China) by mixing yeast/sourdough with flour and water for 10 min to smooth the surface of dough. Next, the dough was divided into 50 g/piece and was placed in an incubator for fermentation at 37°C and 80% relative humidity for 4h [22]. The proofed dough was steamed for 20 min. Each dough treatment was processed in triplicate.”(Line 84 to 106)
Three sourdoughs were collected from each family, please Line 88 to 89.
- Response to comment: (Please make it more explanatory for reproducibility.
Add space between your data and units, except for %. For example, between data and unit of time or temperature
Do this where needed within the manuscript.)
Response: Thank you for the suggestion. The section 2.1 was revised according to the comment, please see Line 91 to 106. Modified throughout the text according to the comment (Line 99).
- Response to comment: (Add the Company's information)
Response: We have added company's information in Line 111 to 112.
- Response to comment: (State the instrument used and model)
Response: We have added the instrument used and model in Line 127 to 128.
- Response to comment: (Model of pH meter)
Response: We have added the model of pH meter in Line 133 to 134.
- Response to comment: (Add references)
Response: We have added citations in Line 129 and Line 134.
- Response to comment: (Add more on sequencing information. For example, primer used and slight procedure.)
Response: Thank you for the suggestion. Line 119 to 120, “The method for the identification of LAB isolates is described by Fujimoto et al. [29].” was added.
- Response to comment: (Be specific on the type of sample collected)
Response: We are grateful for the suggestions. We have modified the sentence according to the comment. Line 136 to 137, the statements of “Samples were collected from the sourdough XS and GS.” were corrected as “The original sourdoughs, collected from XS and GS areas in the selected families, were firstly activated to mature dough (see Section 2.2).”.
- Response to comment: (Is there any addition to this code besides the numbers?)
Response: We have modified the sentence according to the comment. Line 159 to 160, the statements of “LAB Strains 20, 22, 23, 2-2 and 2-7 were grown in MRS broth at 37 °C.” were corrected as “Lactiplantibacillus plantarum 23 and 2-2, Pediococcus pentosaceus 23, 2-7 and Companilactobacillus crustorum 22 were grown in MRS broth at 37 °C.”.
- Response to comment: (How was the steaming done?)
Response: The section 2.5 was revised according to the comment, becoming “ 2.6 Making procedure of steamed bread using LAB sourdoughs
LAB strains 20, 22, 23, 2-2, and 2-7 were grown in MRS broth at 37 °C. Their cells were collected by centrifugation (4 °C, 4192×g, 10 min) and washed twice with sterile water. The cell concentrations of these tested strains were finally adjusted to 1×109 cfu/mL prior to use.
The LAB sourdough was prepared by mixing150 g of wheat flour, 1.5 g of dry yeast, 70 mL of water, and 5 mL of selected bacteria suspensions. The mixture was stirred in a spiral mixer (JHMZ-200, Beijing Oriental Fude Technology Development Co., Ltd. China) for 10 min to smooth the surface of dough. Next, the dough was divided into 50 g/piece and was placed in an incubator for fermentation at 37°C and 80% relative humidity for 4h [22]. The proofed dough was steamed for 20 min. Each dough treatment was processed in triplicate.” (Line 158 to 169).
- Response to comment: (What was the room temperature?)
Response: We have rewrote the section 2.6 according to the comment. The room temperature was be deleted. Please see Line 158 to 169.
- Response to comment: (Write in full before abbreviation for subsequent use)
Response: We really thank for this comment. The full name was be added, please see Line 185 to 187.
- Response to comment: (This is a discussion)
Response: We really thank for this comment. We have rewrote the section according to the reviewer’s comment. We deleted this sentence.
- Response to comment: (The figure did not depict the explanation. A figure or table with level of significant differences can be more appropriate.)
Response: We have revised the figure with level of significant differences according to the suggestion(Line221).
- Response to comment: (Each figure should stand alone with key showing meaning of codes such as WSB etc. The same apply to all tables.)
Response: We have revised it according to the suggestion(Line 241 to 250).
- Response to comment: (This table may not be necessary. Table 3 and 4 have captured the content. The comparison comes in discussion.)
Response: We agree with you. We removed the Table 5.
- Response to comment: (Should appear under discussion)
Response: We really thank for this comment. We deleted this sentence(Line 339).
- Response to comment: (The figure is not well understood. Keys are not clear and units missing)
Response: We agree with you. We reorganized the structure of results. Texture, images and volatile compounds of all bread types was given in one section and compare to each other and the new figure was uploaded, please see Line 212 to 319.
- Response to comment: (What is meant by over-acidified? Duration of fermentation required)
Response: We really thank for this comment. We have rewrote the sentence according to the reviewer’s comment. We deleted this sentence instead of “Lactiplantibacillus plantarum DM616 decreased the hardness of steamed bread after 16 h of fermentation compared with the control.”, please see Line 441 to 442.
- Response to comment: (You mentions studies. References should be more)
Response: We have added citations in Line 533.
- Response to comment: (Recast sentence)
Response: We rephrased this sentence according to the comment (Line 568 to 572).
- Response to comment: (Recast sentence)
Response: We rephrased this sentence according to the comment (Line 582 to 584).
We are trying our best to improve the quality of this manuscript. Some changes take place in the revised manuscript, but these changes will not produce an influence on the content and framework of the presentation. And here we do not list the changes but mark in red in the revised paper.
Finally, we kindly appreciate Editors/Reviewers’ warm work, and hope that this revised version will meet with approval.
Once again, thank you very much for your comments and suggestions.

Reviewer 2 Report
Despite the fact that Chinese steamed bread has been extensively studied, there is always room for improvements in microbial fermentation management. Although the data offered may be interesting, the text needs to be thoroughly revised.
The authors collected three sourdoughs in three different provinces. Indeed not too many, or at least not enough to justify differences between areas.
Authors do not specify whether the collected samples are mature sourdough. As matter of fact, they merely knead these small bits of dough and leave them to ferment for a further 4 h. I'm not sure if this is a typical steamed bread recipe, and in any event, they need to explain why they're doing it to the readers. The section Material and Methods makes no mention of this. However, not-cooked (presumably) samples were subject to textural and image analysis, as well as to VOCs characterization. Such analyses must have been carried out immediately after fermentation, presumably before picking the aliquot for MRS counts (plate coating?? line 109).
Unaccountably, the results of MRS counts are not reported. The authors simply isolated 35 LAB cultures that were identified by 16S rDNA sequencing. Strains were biochemically characterized by API strips, but no findings were provided. Furthermore, tests in MRS were used to characterize strains for their pro-technological ability. To begin with, there is no link between the strains' behavior in MRS and their probable competence in sourdough for steaming bread. the scientists conducted. In addition, tests are so badly described making it difficult to grasp what the authors actually did. At any rate, the results of such tests are not reported as well.
Two of the three samples were then submitted to Illumina sequencing, but no explanation was given for the wx sample's exclusion.
At line 162, the Authors stated that sourdoughs were produced by using some LAB strains, however, they did not specify the screening criteria used. If the recipe for the steamed bread production is the same described in table 1, fermentation would have lasted 4 h, which seems weird. After 4 hours, a LAB strain is still in the Lag phase.
With reference to results, have the impression that outcomes about textural, image and VOCs analyses were reported twice: first for the lone sourdough sample, and again to compare results for the same tests performed on LAB inoculated doughs. In this regard, why the sourdoughs produced by using selected strains were 5 for the textural and image analysis and just four for the VOCs characterization? Strain 22 is missing…
Why did the authors write the whole genome if only the 16S rDNA was sequenced, as shown in tables 3 and 4? Furthermore, 35 strains are insufficient to talk about species distribution or abundance as reported in fig. 4.
It's difficult to get into paragraph 3.5, and at line 290, some strains are classified as dominant strains for no apparent reason.
Illumina sequencing results were presented in six different ways, but I recommend only utilizing a couple. Furthermore, the two sourdoughs were sampled at time 0 and after 4 hours of fermentation, as shown in the figures. I'm curious as to why MRS counts and strains’ isolation were not done at time 0.
Concerning the Discussion, I am confident that integrating this part with the Results would increase the manuscript's clarity. Furthermore, rather than simply describing the available literature, authors should discuss their findings with regard to the current literature on the topic.
In addition, I found two works rather similar to the one here proposed that are not even cited:
Li, Z., Li, H., Deng, C., Bian, K., & Liu, C. (2015). Effect of Lactobacillus Plantarum DM 616 on Dough Fermentation and C hinese Steamed Bread Quality. Journal of Food Processing and Preservation, 39(1), 30-37.
Xi, J., Xu, D., Wu, F., Jin, Z., Yin, Y., & Xu, X. (2020). The aroma compounds of Chinese steamed bread fermented with sourdough and instant dry yeast. Food Bioscience, 38, 100775.
Last but not least, a native speaker must revise the manuscript.
Author Response
Dear Editors and Reviewers,
Thank you very much for your letter with comments on our manuscript entitled “Relationship Between Microbial Composition of Sourdough and Texture, Volatile Compounds of Chinese Steamed Bread” (Foods). These comments are all valuable and helpful for revising our manuscript, and importantly for directing our future researches. We have revised our manuscript according to reviewer’s comments, and would like to re-submit it for your consideration. We have responded the comments raised by the reviewer’s points by points. The amendments are highlighted in red in the revised manuscript. The followings are main corrections in the manuscript and the responds to the reviewer’s comments.
Responds to the reviewer’s comments:
1. Response to comment: (The authors collected three sourdoughs in three different provinces. Indeed not too many, or at least not enough to justify differences between areas.)
Response: We thank the reviewer for the comment. It is really true that the collected sourdoughs are not too many, but we believe the collected samples are very representative. Firstly. the distance between each province exceeds 1000 km. Secondly, there are 13 ethnic groups living in Xinjiang and 16 ethnic groups in Gansu and have developed a specific dietary culture. Thus, we believe the collected samples could support our study. But we will collect more kinds of sourdoughs for research in future work.
2. Response to comment: (Authors do not specify whether the collected samples are mature sourdough. As matter of fact, they merely knead these small bits of dough and leave them to ferment for a further 4 h. I'm not sure if this is a typical steamed bread recipe, and in any event, they need to explain why they're doing it to the readers. The section Material and Methods makes no mention of this. However, not-cooked (presumably) samples were subject to textural and image analysis, as well as to VOCs characterization. Such analyses must have been carried out immediately after fermentation, presumably before picking the aliquot for MRS counts (plate coating?? line 109).)
Response: We really thank for this comment. The collected samples are mature sourdough, the samples of sourdough were activated before be used(Line 98 to 100). We have revised the method of breadmaking to address your concerns and hope that it is now clearer. So, the method has been described as follows: “To evaluate the quality of the products made using sourdough from different provinces, yeast CSB was added as blank control. The recipes of yeast CSB (WSB), Gansu sourdough bread (GSB), and Xinjiang sourdough bread (XSB) are shown in Table 1. The sourdough, basically classified as type I sourdough. Firstly, the samples of sourdough were activated with sourdough and sterile water in the mass ratio of 2:3 and fermented at around 25 ℃ for 9 to 24 h to obtain a mature sourdough. Secondly, dough was prepared as described by Xi et al. (2020) with some modifications (see recipe in Table 1) [29]. Dough was prepared in a spiral mixer (JHMZ-200, Beijing Oriental Fude Technology Development Co., Ltd. China) by mixing yeast/sourdough with flour and water for 10 min to smooth the surface of dough. Next, the dough was divided into 50 g/piece and was placed in an incubator for fermentation at 37°C and 80% relative humidity for 4h [22]. The proofed dough was steamed for 20 min. Each dough treatment was processed in triplicate.” (Line 91 to 106).
In our work, all textural, image analysis and VOCs characterization were analyzed using steamed breads instead of sourdoughs. We have added a flowchart of the experimental design in Line 207 to 210.
3. Response to comment: (Unaccountably, the results of MRS counts are not reported. The authors simply isolated 35 LAB cultures that were identified by 16S rDNA sequencing. Strains were biochemically characterized by API strips, but no findings were provided. Furthermore, tests in MRS were used to characterize strains for their pro-technological ability. To begin with, there is no link between the strains' behavior in MRS and their probable competence in sourdough for steaming bread. the scientists conducted. In addition, tests are so badly described making it difficult to grasp what the authors actually did. At any rate, the results of such tests are not reported as well. Two of the three samples were then submitted to Illumina sequencing, but no explanation was given for the wx sample's exclusion.)
Response: That is a very good question. We apologize for not expressing ourselves clearly. All the experiment including bacterial count test, Gram-stain and 16S rDNA sequencing were be doing to obtain pure isolated. Changes have been made to make the expression clearer and more accurate, please see Line 115 to 116. The purpose of carbohydrate fermentation experiment is to identify dominant strains, and this is a supplemental validation experiment for the results of 16S rDNA sequencing. So, deleted the section "2.3 Phenotype characterization of the optimal LAB The metabolism of several carbohydrates by optimal LAB strains was determined by using API 50 CHL galleries (Marcy-l’Etoile, France) according to the manufacturer instructions. After the incubation period, read the strip according to the bacteria and the type of research. A yellow color indicates a positive reaction and no color change indicates a negative reaction to recorded on the result sheet." to make the expression clearer. Because the sample of WSB as the blank control in textural analysis, Image analysis and volatile compounds analysis. However, we used the Illumina sequencing to
determine the microbiome of sourdough, as blank control, LAB does not exist in the sample of WSB. We reorganized the structure and expressing of methods, please see Line 84 to 134.
4. Response to comment: (At line 162, the Authors stated that sourdoughs were produced by using some LAB strains, however, they did not specify the screening criteria used. If the recipe for the steamed bread production is the same described in table 1, fermentation would have lasted 4 h, which seems weird. After 4 hours, a LAB strain is still in the Lag phase.)
Response: Thank you for underlining this deficiency. We choose the strains with fast growth and high acid production as the optimal strains. The method of screening for optimal LAB, please see Line 124 to 134. In the sections Materials and Methods, we have rewrote the method of steamed breadmaking. All the sourdough was activated to obtain a mature sourdough before being used. Thus, 4h is enough for growth of LAB. Please see Line 92 to 106.
5. Response to comment: (With reference to results, have the impression that outcomes about textural, image and VOCs analyses were reported twice: first for the lone sourdough sample, and again to compare results for the same tests performed on LAB inoculated doughs. In this regard, why the sourdoughs produced by using selected strains were 5 for the textural and image analysis and just four for the VOCs characterization? Strain 22 is missing…)
Response: We really thank for this comment. We reorganized the structure of results. Texture, images and volatile compounds of all bread types was given in one section and compare to each other, please see Line 212 to 319. We selected Lactiplantibacillus plantarum 20, 2-2 and Pediococcus pentosaceus 23, 2-7 as a pairwise comparison to evaluate the effect of volatile compounds of different LAB. We did not isolate Companilactobacillus crustorum from Gansu sourdough.
6. Response to comment: (Why did the authors write the whole genome if only the 16S rDNA was sequenced, as shown in tables 3 and 4? Furthermore, 35 strains are insufficient to talk about species distribution or abundance as reported in fig. 4.)
Response: Thank you for your suggestion. We have added a new flowchart of the experimental design to further illustrate(Line 207 to 210). The 16S rDNA sequencing were be done to obtain pure isolated. For talk about the species distribution or abundance of sourdough, we used Illumina high-throughput sequencing to investigation.
7. Response to comment: (It's difficult to get into paragraph 3.5, and at line 290, some strains are classified as dominant strains for no apparent reason.)
Response: We really thank for this comment. We have modified the sentence according to the previous comment. We choose the strains with fast growth and high acid production as the optimal strains. We have rewrote the section as follows: “Analysis of the growth performance of the isolated sourdough LAB in GS at different times (Figure S1) revealed that the 2-2 strain had the best growth ability, and 2-1 strain also had good growth ability. With respect to the ability of selected LAB to grow under acidic environments, strain 2-2, 2-7 had the most rapid decrease in pH (Figure S2). Therefore, strain 2-2 is a fast-growing and acid-producing strain in GS. Analysis of the growth performance of the isolated sour-dough LAB in XS at different times (Figure S3) revealed that strain 18, 21, displayed great growth ability. With respect to the ability of selected LAB to grow under acidic environments, strain 20, 23 showed the most rapid decrease in pH (Figure S4). As shown in Figure S1, Figure S2, Figure S3 and Figure S4, Lactiplantibacillus plantarum 2-2, Lactiplantibacillus plantarum 2-1, and Pediococcus pentosaceus 2-7 are dominant strains in the sourdough GS, while strains Lactiplantibacillus plantarum 3, Lactiplantibacillus plantarum 18, Lactiplantibacillus
plantarum 20, and Pediococcus pentosaceus 23dominate the sourdough XS. Clearly, Lactiplantibacillus plantarum 23and 2-2, as observed in their growth capacity and pH determination experiments, should be selected as the dominant strain of the sourdoughs, and Pediococcus pentosaceus 23, 2-7 is another dominate species responsible for the dough acidification.”(Line 341 to 356)
8. Response to comment: (Illumina sequencing results were presented in six different ways, but I recommend only utilizing a couple. Furthermore, the two sourdoughs were sampled at time 0 and after 4 hours of fermentation, as shown in the figures. I'm curious as to why MRS counts and strains’ isolation were not done at time 0.)
Response: We agree the interesting comments, however, due to better description of sequencing results, we believe that it's necessary to describe by microbial community diversity analysis, alpha diversity and beta diversity analysis. The bacterial count test and strains’ isolation were done using the mature sourdough at time 0. In order to explore the dynamic changes of LAB during the fermentation of sourdough, we sampled at time 0h and 4h, respectively.
9. Response to comment: (Concerning the Discussion, I am confident that integrating this part with the Results would increase the manuscript's clarity. Furthermore, rather than simply describing the available literature, authors should discuss their findings with regard to the current literature on the topic. In addition, I found two works rather similar to the one here proposed that are not even cited:
Li, Z., Li, H., Deng, C., Bian, K., & Liu, C. (2015). Effect of Lactobacillus Plantarum DM 616 on Dough Fermentation and C hinese Steamed Bread Quality. Journal of Food Processing and Preservation, 39(1), 30-37.
Xi, J., Xu, D., Wu, F., Jin, Z., Yin, Y., & Xu, X. (2020). The aroma compounds of Chinese steamed bread fermented with sourdough and instant dry yeast. Food Bioscience, 38, 100775.)
Response: Thank you for providing these insights. But we need to follow the formatting requirements of manuscript in foods. Discussion was revised and modified according to Reviewer’s suggestion, Line 441 to 442, “Lactiplantibacillus plantarum DM616 decreased the hardness of steamed bread after 16 h of fermentation compared with the control[31].” was added, Line 554 to 556, “Xi' finding showed that some important aroma compounds found in both CSB were the same, they were significantly different in their concentrations[28].” was added. We have added citation in Line 441 to 442 and Line 554 to 556.
11. Response to comment: (Last but not least, a native speaker must revise the manuscript.)
Response: We really thank for this comment. Our manuscript undergone extensive English editing services by MDPI.
We are trying our best to improve the quality of this manuscript. Some changes take place in the revised manuscript, but these changes will not produce an influence on the content and framework of the presentation. And here we do not list the changes but mark in red in the revised paper.
Finally, we kindly appreciate Editors/Reviewers’ warm work, and hope that this revised version will meet with approval. Once again, thank you very much for your comments and suggestions.

Reviewer 3 Report
There many lack of information related to sourdough in the manuscript although the authors did good work. They should have learned the basic knowledge regarding sourdough. They don’t know which type of sourdough they used (e.g Type I or Type II) and this is the important point of the sourdough studies. They should focus on these points and mention which type of sourdough they used in the whole manuscript . Additionally, the study contains so many data and is not easy to follow.
- Introduction
1.1 The first sentences of introduction (Line 29) is not completely true. Although there are many definitions of sourdough, a widely accepted definition was made by Gobbetti (1998).
The authors revise the first sentences according to the reference which stated at below.
Ref: Arora, K., Ameur, H., Polo, A., Di Cagno, R., Rizzello, C. G., & Gobbetti, M. (2021). Thirty years of knowledge on sourdough fermentation: A systematic review. Trends in Food Science & Technology, 108, 71-83.
1.2 The authors should revise this sentence (Line 35-37) “Data from several studies suggest that the production of most volatile substances are related to the metabolism and enzyme conversion by LAB, and volatile substances in connection with the fermentation types of LAB” because it is not clear. Also, the reference they cited (number 4) seems to be irrelevant, it should be checked and revised.
1.3 In line 40, what is the meaning of “fermentation types of LAB”? It should be clarified.
1.4 In line 43, the term of “microorganisms “is better instead of “microbes”.
1.5 In line 45, it should be surface not skin.
1.6 In line 47, not only for Chinese steamed breads, it is valid for all sourdough bread. So, the sentence should be revised.
1.7 In line 49 the sentence starting with face tough…….should be revised.
1.8 Similarly, the sentence in line 50 (Hand-made products……..) should be revised, it is not clear.
1.9 The sentence in line 54 to 56 is also not clear.
1.10 The sentence in line 56-58 is detached from the main text.
1.11 The sentences in line 64-69 contain irrelevant knowledge independent from the manuscript. It should be omitted.
1.12 Last paragraph of the introduction must be revised because it is not clear.
Overall, the introduction part must be revised because it is difficult to follow and there are so many grammatically incorrect sentences.
- Materials and Methods
- Last paragraph of the section 2.1 should be separated from this part and mentioned in a new section because it is not related to sourdough preparation and sampling.
- How many types of bread are there? It is so confused because it mentions LAB sourdoughs fermentation in section 2.5. Did the authors produce a steamed bread with type I sourdough or type II sourdough or both of them? It must be mentioned and clarify.
- The reference of section 2.7 should be given.
- This section is also difficult to flow. Maybe, it is a good idea to give a flowchart of the experimental design.
- Results
- It should be reorganised. Texture of all bread types should be given in one section and compare to each other. Similarly, images of all breads types should be merged and also volatile compounds.
- Related to Figure 3, authors should give more information about the PCA.
- Discussion
- Line 444: “The current study found that over-acidified sourdough products were harder and slightly acidified sourdough products were softer.” How did the authors come to this idea? Because, no pH measurement were performed for the bread dough or sourdough. So, How do you know the acidity of the products?
- Similarly, in line 535, “The current study found that pH of sourdoughs with the addition of LAB cultures decreased obviously comparing to the 536 dough without LAB.” There is no information how the pH of sourdough was measured in method section.
- Conclusion
- Sourdough, not sour dough, it should be corrected whole manuscript.
- In line 584 what is the meaning of “old sourdough” It should be traditional sourdough.

Author Response
Dear Editors and Reviewers,
Thank you very much for your letter with comments on our manuscript entitled “Relationship Between Microbial Composition of Sourdough and Texture, Volatile Compounds of Chinese Steamed Bread” (Foods). These comments are all valuable and helpful for revising our manuscript, and importantly for directing our future researches. We have revised our manuscript according to reviewer’s comments, and would like to re-submit it for your consideration. We have responded the comments raised by the reviewer’s points by points. The amendments are highlighted in red in the revised manuscript. The followings are main corrections in the manuscript and the responds to the reviewer’s comments.
Responds to the reviewer’s comments:
1. Response to comment: (The first sentences of introduction (Line 29) is not completely true. Although there are many definitions of sourdough, a widely accepted definition was made by Gobbetti (1998). The authors revise the first sentences according to the reference which stated at below. Ref: Arora, K., Ameur, H., Polo, A., Di Cagno, R., Rizzello, C. G., & Gobbetti, M. (2021). Thirty years of knowledge on sourdough fermentation: A systematic review. Trends in Food Science & Technology, 108, 71-83.)
Response: We really thank you for this comment. We have modified the sentence according to the comment (see Line 30 to 32). The statements of “Sourdough is a traditional dough starter, which is generally fermented with water, grain and microorganisms.” were corrected as “Sourdough is a traditional dough starter. It is a mixture of flour and water, spontaneously fermented by lactic acid bacteria and yeasts, with acidification and fermentable capacities[1,2].”.
2. Response to comment: (The authors should revise this sentence (Line 35-37) “Data from several studies suggest that the production of most volatile substances are related to the metabolism and enzyme conversion by LAB, and volatile substances in connection with the fermentation types of LAB” because it is not clear. Also, the reference they cited (number 4) seems to be irrelevant, it should be checked and revised.)
Response: We have modified the sentence according to the comment. Line 37 to 38, the statements of “Data from several studies suggest that the production of most volatile substances are related to the metabolism and enzyme conversion by LAB, and volatile substances in connection with the fermentation types of LAB[4].” were corrected as “Data from several studies suggest that the differences in volatile substances were mainly attributed to the metabolism and enzyme conversion by different LAB[9,10].”. We also have replaced new citation in Line 38.
3. Response to comment: (In line 40, what is the meaning of “fermentation types of LAB”? It should be clarified.)
Response: We have revised the text to address your concerns and hope that it is now clearer. We have modified the sentence according to the comment. Line 40 to 42, the statements of “Due to the differences between metabolites and fermentation types of LAB, the volatile substances produced by LAB are species-specific[10].” were corrected as “Homo- and heterofermentative metabolism of LAB differ with respect to the of flavor formation in bread and enable the volatile substances produced by LAB, which are species-specific.[13–16].”.
4. Response to comment: (In line 43, the term of “microorganisms “is better instead of “microbes”.)
Response: We have changed it according to the suggestion(Line 44).
5. Response to comment: (In line 45, it should be surface not skin.)
Response: We have changed it according to the suggestion(Line 47).
6. Response to comment: (In line 47, not only for Chinese steamed breads, it is valid for all sourdough bread. So, the sentence should be revised.)
Response: Thank you for the suggestion, We have modified the sentence according to the comment. Line 48 to 50, the statements of “Thus, selected specific microbiota as starter cultures plays a vital role in the developing of Chinese steamed breads with specific flavor properties and quality.” were corrected as “Thus, selected specific microbiota as starter cultures play a vital role in the development of sourdough breads with specific flavor properties and quality, especially in the slow industrialization of Chinese steamed bread.”.
7-11. Response to comment: (7. In line 49 the sentence starting with face
tough…….should be revised.
8. Similarly, the sentence in line 50 (Hand-made products……..) should be revised, it is not clear.
9. The sentence in line 54 to 56 is also not clear.
10. The sentence in line 56-58 is detached from the main text.
11. The sentences in line 64-69 contain irrelevant knowledge independent from the manuscript. It should be omitted.)
Response: We are grateful for the suggestions. To be more clear and in accordance with the reviewer concerns, we have rewrote the introduction as follows: “Sourdough is a product with regional characteristics. It is a unique food ecosystem in that it selects LAB that are adapted to their environment and harbors dominant species for sourdough[5]. Several studies analyzing the microbial communities in sourdough collected from different regions of China have shown dramatic differences in the dominant strains[21,22]. Leuconostoc citreum HO12 and Weissella koreensis HO20 isolated from kimchi were assessed as starter cultures in the making of sourdough bread[16]. Four LAB strains were tested for the preparation and propagation of sourdoughs for the production of a typical bread at the industrial level[23]. It was confirmed that dominant strains apparently played a significant role in benefiting the quality of sourdough, improving the mouth-feel of bread. A clear understanding of the interaction between the dominant strains of sourdoughs and the quality of steamed breads will be beneficial for practice. Thus, it is essential to identify the composition of the dominant strains in the sourdough, selecting the dominant strains as starter cul-tures for promoting the quality of Chinese steamed bread.
Chinese steamed bread is a widely consumed food, representing about 40% of the wheat consumption in China, especially in Northwestern China[24]. Local people in Northwestern China including Gansu, Xinjiang, and Ningxia, Qinghai and Shaanxi areas where high-quality and high-yielding wheat is produced, almost enjoy sourdough-based steamed buns every day [26]. Especially, the Uyghurs and Hui Minorities have preserved their own dietary habits and specific lifestyles, developing a specific dietary culture in Northwestern China over time[25]. A recent research has demonstrated the possible influence of dietary habits, geographic location and ethnicity on the gut microbiota[26]. However, few reports dealing with the linking of sourdoughs from Northwestern China to their steamed-bread quality have been presented[21,27].” (Line 51 to 73)
12. Response to comment: (Last paragraph of the introduction must be revised because it is not clear.)
Response: Thank you for underlining this deficiency. This section was revised and modified according to the comment, becoming “The objective of this study was to investigate the relationship between the microbial communities of sourdough obtained from Xinjiang and Gansu areas and the texture and volatile compounds of corresponding Chinese steamed breads. This work is trying to elucidate the connection of the dominant LAB strains with the traditional Chinese steamed bread's quality through adjusting microorganisms in sourdough.”. (Line 78 to 82)
13. Response to comment: (Overall, the introduction part must be revised because it is difficult to follow and there are so many grammatically incorrect sentences.)
Response: We really thank for this comment. We have exhaustively revised the introduction according to the comment. Our manuscript undergone extensive English editing services by MDPI.
14. Response to comment: (Last paragraph of the section 2.1 should be separated from this part and mentioned in a new section because it is not related to sourdough preparation and sampling.)
Response: We are grateful for the suggestions. The section 2.1 had be separated from this part and mentioned in a new section as follows: “
2.1 Sourdough sampling
The sourdoughs analyzed in this study were collected from different families lo-cated in the Provinces of Xinjiang (XS) and Gansu (GS) of China, which produce sour-dough bread without any added salt. The selected families have been making Chinese steamed bread (CSB) with sourdough every day for more than 10 years. Three sour-doughs were collected from each family. All samples were kept at low temperature (4 °C) during transportation and storage.
2.2 Chinese steamed breadmaking using sourdough
To evaluate the quality of the steamed-breads made using sourdoughs from different areas, yeast CSB was added as blank control. The recipes of yeast CSB (WSB), Gansu sourdough bread (GSB), and Xinjiang sourdough bread (XSB) are shown in Table 1. Firstly, the sourdough, basically classified as type I sourdough, was activated with sourdough and sterile water in the mass ratio of 2:3 and fermented at 25 ℃ for 9 to 24 h to obtain a mature sourdough. Secondly, dough was prepared as described by Xi et al. (2020) with some modifications (see recipe in Table 1)[28]. Dough was prepared in a spiral mixer (JHMZ-200, Beijing Oriental Fude Technology Development Co., Ltd. China) by mixing yeast/sourdough with flour and water for 10 min to smooth the surface of dough. Next, the dough was divided into 50 g/piece and was placed in an incubator for fermentation at 37 °C with 80% relative humidity for 4h[22]. The proofed dough was steamed for 20 min. Each dough treatment was processed in triplicate.”(Line 84 to 106)
15. Response to comment: (How many types of bread are there? It is so confused because it mentions LAB sourdoughs fermentation in section 2.5. Did the authors produce a steamed bread with type I sourdough or type II sourdough or both of them? It must be mentioned and clarify.)
Response: There are 7 kinds of bread, including WSB, XSB, GSB and 4 kinds of LAB sourdough bread. The section 2.5 was revised according to the comment, becoming “ 2.6 Making procedure of steamed bread using LAB sourdoughs Lactiplantibacillus plantarum 23 and 2-2, Pediococcus pentosaceus 23, 2-7 and Companilactobacillus crustorum 22 were grown in MRS broth at 37 °C. Their cells were collected by centrifugation (4 °C, 4192×g, 10 min) and washed twice with sterile water. The cell concentrations of these tested strains were finally adjusted to 1×109 cfu/mL prior to use.
The LAB strains-based sourdoughs were prepared by mixing150 g of wheat flour, 1.5 g of dry yeast, 70 mL of water, and 5 mL of selected bacteria suspensions. The mixture was stirred in a spiral mixer (JHMZ-200, Beijing Oriental Fude Technology Development Co., Ltd. China) for 10 min to smooth the surface of dough. Next, the dough was divided into 50 g/piece and was placed in an incubator for fermentation at 37°C with 80% relative humidity for 4h [22]. The proofed dough was steamed for 20 min. Each dough treatment was processed in triplicate.” (Line 158 to 169). The sourdough was type I sourdough, please see Line 98.
16. Response to comment: (The reference of section 2.7 should be given.)
Response: We have added citation in Line 171.
17. Response to comment: (This section is also difficult to flow. Maybe, it is a good idea to give a flowchart of the experimental design.)
Response: We thank the reviewer for the very significant comment. We have added a flowchart of the experimental design in Line 207 to 210.
18. Response to comment: (It should be reorganised. Texture of all bread types should be given in one section and compare to each other. Similarly, images of all breads types should be merged and also volatile compounds.)
Response: We are grateful for the suggestions. We reorganized the structure of results. Texture, images and volatile compounds of all bread types was given in one section and compare to each other, please see Line 212 to 319.
19. Response to comment: (Related to Figure 3, authors should give more information about the PCA.)
Response: We have added the information about the PCA of volatile compounds of all bread (Lines 284 to 290 and Lines 308 to 319).
20. Response to comment: (Line 444: “The current study found that over-acidified sourdough products were harder and slightly acidified sourdough products were softer.” How did the authors come to this idea? Because, no pH measurement were performed for the bread dough or sourdough. So, How do you know the acidity of the products?)
Response: Thank you for the suggestion, we have modified the sentence according to the comment. We removed this sentence and added a new citation. Line 441 to 442, the statements of “The current study found that over-acidified sourdough products were harder and slightly acidified sourdough products were softer[42,43].” were corrected as
“Lactiplantibacillus plantarum DM616 decreased the hardness of steamed bread after 16 h of fermentation compared with the control[31].”.
21. Response to comment: (Similarly, in line 535, “The current study found that pH of sourdoughs with the addition of LAB cultures decreased obviously comparing to the 536 dough without LAB.” There is no information how the pH of sourdough was measured in method section.)
Response: We have modified the sentence according to the comment. Line 534 to 537, the statements of “The current study found that pH of sourdoughs with the addition of LAB cultures de-creased obviously comparing to the dough without LAB. Sourdoughs with various LAB showed different range of pH and total titratable acidity (TTA) [69].” were corrected as “Hadaegh et al. found that pH of sourdoughs with the addition of LAB cultures decreased obviously compared to the dough without LAB, and sourdoughs with various LAB showed different ranges of pH and total titratable acidity (TTA) [73].”. This sentence is a citation, and the cited literature measures the TTA of sourdough.
22. Response to comment: (Sourdough, not sour dough, it should be corrected whole manuscript.)
Response: We have modified this expression throughout the text according to the comment. Please see Line 409 and 560.
23. Response to comment: (In line 584 what is the meaning of “old sourdough” It should be traditional sourdough.)
Response: We have changed it according to the suggestion(Line 582).
We are trying our best to improve the quality of this manuscript. Some changes take place in the revised manuscript, but these changes will not produce an influence on the content and framework of the presentation. And here we do not list the changes but mark in red in the revised paper.
Finally, we kindly appreciate Editors/Reviewers’ warm work, and hope that this revised version will meet with approval.
Once again, thank you very much for your comments and suggestions.

Round 2
Reviewer 3 Report
Although the authors revised the manuscript by considering the comments, there are still some minor points that should be revised.

Author Response
Dear Editors and Reviewers,
Thank you very much for your letter with comments on our manuscript entitled “Relationship Between Microbial Composition of Sourdough and Texture, Volatile Compounds of Chinese Steamed Bread” (Foods). Firstly, we are very appreciating for giving us second opportunity to revise our manuscript. Secondly, these comments are all valuable and helpful for revising and improving our manuscript, as well as the important guiding significance to our researches.
We have revised our manuscript according to reviewer’s comments, and would like to re-submit it for your consideration. We have addressed the comments raised by the reviewers, and the amendments are highlighted in red in the revised manuscript. Point by point responses to the reviewers’ comments are listed below.The followings are main corrections in the manuscript and the responds to the reviewer’s comments.
Responds to the reviewer’s comments:
- Response to comment: (In line 41 : ‘’Homo- and heterofermentative metabolism of LAB differ with respect to the of flavor formation in bread and enable the volatile substances produced by LAB, which are species-specific’’ In this sentences of flavor should be off-flavor or only the flavor? It should be clarify.)
Response: Thank you for the suggestion. We have modified the sentence according to the comment. Line 40 to 42, the statements of “Homo- and heterofermentative metabolism of LAB differ with respect to the of flavor formation in bread and enable the volatile substances produced by LAB, which are species-specific.” were corrected as “Homo- and heterofermentative metabolism of LAB differ with respect to the flavor formation in bread and enable the volatile substances produced by LAB, which are species-specific.”.
- Response to comment: (In line 51 : ‘’Sourdough is a product with regional characteristics.’’ With should be change with having.)
Response: We really thank for this comment. We have modified the sentence according to the comment. Line 51, the statements of “Sourdough is a product with regional characteristics.” were corrected as “Sourdough is a product having regional characteristics.”.
- Response to comment: (In line 53 : “for sourdough’’ should be omitted.)
Response: We have omitted it according to the suggestion(Line 51 to 53).
- Response to comment: (In line 61,62 and 63 : Thus, it is essential to identify the composition of the dominant strains in the sourdough, selecting the dominant strains as starter cultures for promoting the quality of Chinese steamed bread. The composition of the dominant strains in the sourdough should be revised as the composition sourdough.)
Response: We are grateful for the suggestions. We have modified the sentence according to the comment. Line 61 to 63, the statements of “Thus, it is essential to identify the composition of the dominant strains in the sourdough, selecting the dominant strains as starter cultures for promoting the quality of Chinese steamed bread.” were corrected as “Thus, it is essential to identify the composition of sourdough, selecting the dominant strains as starter cultures for promoting the quality of Chinese steamed bread.”.
- Response to comment: (In line 70 and 71: I definitely think this sentence should be removed (A recent research has demonstrated the possible in- 70 fluence of dietary habits, geographic location and ethnicity on the gut microbiota) Because it is irrelevant.)
Response: Thank you for the suggestion. We deleted this sentence. Please see Line 70.
- Response to comment: (For Table 1 : The meaning of WS, GS and XS should be given below the table.)
Response: We have added the note below the Table 1. “Note: The 3 types of bread are as follows, yeast bread (WSB), Gansu sourdough bread (GSB), Xinjiang sourdough bread (XSB).” was be added. Please see Line 96 to 97.
We are trying our best to improve the quality of this manuscript. Some changes take place in the revised manuscript, but these changes will not produce an influence on the content and framework of the presentation.
Finally, we kindly appreciate Editors/Reviewers’ warm work, and wish that the revised version of the manuscript is now acceptable for publication in your journal.
Once again, thank you very much for your comments and suggestions.
